# Counterfactuals As a Means for Evaluating Faithfulness of Attribution Methods in Autoregressive Language Models

**Sepehr Kamahi[1], Yadollah Yaghoobzadeh[1,2]**
[1]School of Electrical and Computer Engineering
College of Engineering, University of Tehran, Tehran, Iran
[2]Tehran Institute for Advanced Studies, Khatam University, Tehran, Iran
sepehr.kamahi@ut.ac.ir, y.yaghoobzadeh@ut.ac.ir

## Abstract

Despite the widespread adoption of autoregressive language models, explainability evaluation research has predominantly focused on span infilling and masked language models. Evaluating the faithfulness of an explanation method—how accurately it explains the inner workings and decision-making of the model—is challenging because it is difficult to separate the model from its explanation. Most faithfulness evaluation techniques corrupt or remove input tokens deemed important by a particular attribution (feature importance) method and observe the resulting change in the model's output. However, for autoregressive language models, this approach creates out-of-distribution inputs due to their next-token prediction training objective. In this study, we propose a technique that leverages counterfactual generation to evaluate the faithfulness of attribution methods for autoregressive language models. Our technique generates fluent, in-distribution counterfactuals, making the evaluation protocol more reliable.

## 1 Introduction

Most modern NLP systems rely on autoregressive, transformer-based language models (Brown et al., 2020; Touvron et al., 2023; Groeneveld et al., 2024). These models are inherently opaque, creating a strong need to understand their decision-making processes. As a result, explanation methods have become increasingly important in the field.

A widely-used approach for model explainability is attribution, also known as feature importance (FI) (Zhao et al., 2023). Attribution methods aim to identify which input features contribute most to a model's predictions, assigning a scalar value to each feature that reflects its relevance in the decision-making process. In typical NLP tasks, input features are often subwords or their combinations.

A key challenge in evaluating the faithfulness of attribution methods is that many existing techniques are designed for denoising or masked language models (MLMs) (Kobayashi et al., 2020, 2021; Ferrando et al., 2022b; Modarressi et al., 2022, 2023; Mohebbi et al., 2023). Recent work on autoregressive models has primarily focused on the plausibility of attributions (Yin and Neubig, 2022; Ferrando et al., 2023). While plausible (or persuasive) explanations might be the objective of the explainer, the core objective for the user is to truly understand the model's decision-making process, rather than simply being convinced that the model's decisions are correct (Jacovi and Goldberg, 2021).

Nearly all previous methods for faithfulness evaluation modify the input in some way, such as masking or removing important tokens based on the attribution results, and then measuring the impact on the model's predictions. These methods tend to work well for MLMs, which are specifically trained for tasks like span or mask infilling. However, in the case of autoregressive models like GPT-2, which predict the next token, such modifications produce out-of-distribution (OOD) inputs. This raises a crucial question: are these evaluation methods truly assessing the informativeness of the selected tokens, or merely testing the model's robustness to unnatural text and the artifacts introduced by testing modifications (Hooker et al., 2019)? Moreover, the OOD nature of these inputs results in explanations that become socially misaligned (Hase et al., 2021). In other words, the expectations of users—who seek to understand which features are most relevant to the model's decision—no longer align with the actual output of the attribution method. Instead, feature importance becomes influenced by the model's priors rather than the learned features that truly drive predictions.

In this work, drawing inspiration from coun-

terfactual generation—where the input is altered to flip the model's output—we propose a new technique to evaluate the faithfulness of attribution methods in autoregressive language models. Specifically, we use counterfactual generators to modify the input by focusing on tokens highlighted by attribution methods, while ensuring that the altered input remains natural, fluent, and within the model's original distribution. This ensures that any observed change in the model's predictions is due to the modification of the important tokens, rather than an effect of OOD inputs.

We argue that if an attribution method enables a counterfactual generator to modify fewer tokens to change the model's prediction, then it demonstrates a stronger understanding of the model's inner workings, indicating higher faithfulness. To validate our approach, we apply this faithfulness evaluation technique to several attribution methods—including gradient norm, gradient × input, erasure, KernelSHAP, and integrated gradients—within the context of next-word prediction for two language models: the fine-tuned Gemma-2b and the off-the-shelf Gemma-2b-instruct (Team et al., 2024).

Our contributions are as follows: (i) We introduce a novel faithfulness evaluation protocol that preserves the model's input distribution, designed for attribution methods in autoregressive language models. (ii) We apply this protocol to evaluate and rank widely-used attribution methods, showcasing differences in sensitivity between fine-tuned and off-the-shelf models when handling OOD data and proposing a solution.[1]

## 2 Related work

**Evaluating Explanations.** Most current metrics for evaluating faithfulness involve either removing important tokens or retraining the model using only those identified as important by attribution methods (Chan et al., 2022). For instance, Abnar and Zuidema (2020) assess explanations by comparing them with gradient and ablation techniques. Although Wiegreffe and Pinter (2019) caution that gradients should not be considered ideal or the "ground truth," they still utilize gradients as a proxy for the model's intrinsic semantics. Importantly, the trustworthiness of explanations is both task- and model-dependent (Bastings et al., 2022), and

---

[1]The code is available at https://github.com/Sepehr-Kamahi/faith

different attribution methods frequently produce inconsistent results (Neely et al., 2022). As a result, it is not justifiable to treat any single explanation method as a universal standard across all contexts.

In their work, DeYoung et al. (2020) introduce two key concepts: comprehensiveness (whether the important tokens identified are the only ones necessary for making a prediction) and sufficiency (whether these important tokens alone are enough to make the prediction). Carton et al. (2020) build on this by proposing normalized versions of these concepts, comparing comprehensiveness and sufficiency to the null difference—the performance of an empty input (for sufficiency) or a full input (for comprehensiveness). However, it remains unclear whether these corruption techniques evaluate the informativeness of the corrupted tokens or merely the robustness of the model to unnatural inputs and artifacts introduced during evaluation.

Further, Han et al. (2020) and Jain et al. (2020) frame attribution methods as either faithful or unfaithful, with no consideration for degrees of faithfulness. They describe attribution methods that are "faithful by construction." In contrast, other researchers propose that faithfulness exists on a spectrum and suggest evaluating the "degree of faithfulness" of explanation methods (Jacovi and Goldberg, 2020). Our approach aligns with this view, as we aim to find explanation methods that are sufficiently faithful for autoregressive models.

Atanasova et al. (2023) evaluate the faithfulness of natural language explanations using counterfactuals, applying techniques from Ross et al. (2021) to assess how well explanations align with the model's decision-making. This line of work offers valuable insights into the use of counterfactuals, which we build upon for evaluating attribution methods in language models. Another relevant direction is the evaluation of explanations using uncertainty estimation. For example, Slack et al. (2021) develop a Bayesian framework that generates feature importance estimates along with their associated uncertainty, expressed through credible intervals, highlighting the importance of uncertainty in faithfulness evaluations

**The OOD Problem in Explainability.**

The issue of OOD inputs in explainability has been raised by several works. Hooker et al. (2019) and Vafa et al. (2021) suggest retraining or fine-tuning the model using partially erased inputs to align training and evaluation distributions. However, this process can be computationally expen-

sive and is not always practical. An alternative approach by Kim et al. (2020) aims to ensure that the explanation remains in-distribution to mitigate OOD problems. Our work addresses this concern by preserving the input distribution during faithfulness evaluation, particularly for autoregressive models.

**Feature Importance (Attribution).** Attributions, or feature importance scores, are local explanations that assign a score to each input feature—typically token embeddings in NLP tasks—indicating how crucial that feature is to the model's prediction. Attribution methods can be categorized into four types: i) Perturbation-based methods, which alter or mask input features to assess their importance by observing changes in the model's output (Li et al., 2016, 2017; Feng et al., 2018; Wu et al., 2020). ii) Gradient-based methods, which calculate the derivative of the model's output with respect to each input to measure the influence of each feature (Mohebbi et al., 2021; Kindermans et al., 2019; Sundararajan et al., 2017; Lundstrom et al., 2022; Enguehard, 2023; Sanyal and Ren, 2021; Sikdar et al., 2021). iii) Surrogate-based methods, which explain a complex black-box model using a simpler, interpretable model (Ribeiro et al., 2016; Lundberg and Lee, 2017; Kokalj et al., 2021). iv) Decomposition-based methods, which break down the overall importance score into linear contributions from the input features (Montavon et al., 2019; Voita et al., 2021; Chefer et al., 2021; Modarressi et al., 2022; Ferrando et al., 2022a).

## 3 Our method

Our faithfulness evaluation protocol involves two models: a counterfactual generator model and a predictor model. Our goal is to evaluate the faithfulness of attribution methods for the predictor model. Due to the large output space of autoregressive language models (LMs), which often includes thousands of vocabulary items, examining the entire output space does not provide much insight. Therefore, we use the contrastive explanations proposed by Yin and Neubig (2022), which measure the attribution of input tokens for a contrastive model decision. Contrastive attributions aim to identify the most important tokens that led the model to predict the target $y_t$ instead of a foil $y_f$. We then use a separate editor model to modify these important tokens to generate counterfactuals—examples that make the original predictor model more likely to

Figure 1: Prompting techniques used for counterfactual generation in the second phase.

predict the foil.

Our protocol for evaluating attributions consists of two phases. The first phase involves creating the editor that can generate counterfactuals. In the second phase, we use the editor and predictor together to determine what percentage of tokens the editor needs to change to flip the predictor model's prediction. Figure 2 illustrates the second phase.

To create the editor, we fine-tune an autoregressive language model specifically for counterfactual generation. During fine-tuning, we add two tokens to the embedding space and the tokenizer: '<mask>' and '<counterfactual>'. Inspired by Wu et al. (2021) and Donahue et al. (2020), we create training examples for our counterfactual generator by randomly masking between 5% and 50% of the tokens. We then append each example's label (e.g., positive or negative for the SST-2 dataset), the '<counterfactual>' token, and finally the original unmasked example. The process of creating training examples is shown in Figure 3.

In the second phase of evaluating attributions, we first input a sentence into the predictor and apply an attribution method to identify the most important tokens influencing the predictor's decision-making process. We begin by replacing 10% of these most important tokens with '<mask>' and present the masked sentence along with the foil label (the label with the second-highest logit) to the *editor* to generate a counterfactual sentence—one that flips the prediction of the predictor model. If unsuccessful in flipping the prediction, we incrementally increase the masking by 10% until we either flip the prediction or reach a masking threshold of 50%. This evaluation protocol is depicted in Figure 2. The prompting technique used for counterfactual generation during this phase is shown in Figure 1. The attribution technique that identifies the most critical tokens for creating counterfactuals and enables counterfactuals with the least amount of change to the original text is considered to provide the most faithful representation of the predictor's decision-making process.

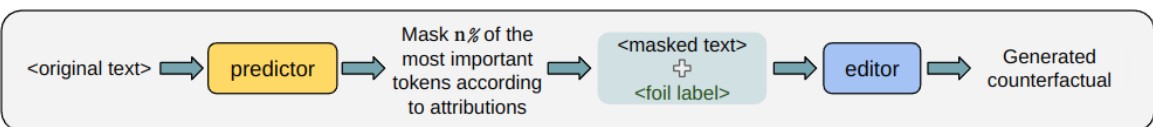

Figure 2: Our process of generating counterfactuals for evaluating attribution methods. The predictor (an LM) generates a label for the given text, and an attribution method specifies the most important tokens. We mask the top n% of them and ask an editor (another LM) to change the label of the input text by filling in the masked tokens. If the attribution method is more faithful, then the required n% should be lower.

```
<randomly masked text> <original label><counterfactual><unmasked text>
Example:
<mask> a clear<mask> of<mask> positive<counterfactual> a clear sense of purpose
```

Figure 3: Creation of training examples for fine-tuning the counterfactual generator, and one given sample.

## 4 Experimental Setup

### 4.1 Datasets

We use three datasets for evaluating faithfulness: SST-2 (Socher et al., 2013) and IMDB (Maas et al., 2011), which are both binary classification datasets, and AG-News (Zhang et al., 2015), a four-class classification dataset.

Faithfulness evaluation datasets should not have gold attribution labels because we do not want human intuition to influence the evaluation. Instead, we aim to understand how the model makes predictions (Jacovi and Goldberg, 2020).

### 4.2 Models

#### 4.2.1 Editor Models

For the editor model, our method is similar to Wu et al. (2021), which uses GPT-2, a decoder-only causal model, for generating counterfactuals. We extend this by using three more modern decoder-only models: GPT-J-6B (Wang and Komatsuzaki, 2021), which we refer to as "gptj," and two sizes of Pythia: Pythia-1.4B (pythia1) and Pythia-2.8B (pythia2) (Biderman et al., 2023). We fine-tune these models following the process described in Section 3. The pythia1 model is fully fine-tuned, while the other two (gptj and pythia2) are fine-tuned using Low-Rank Adaptation (LoRA) (Hu et al., 2022). All models are trained for 8 epochs using dynamic masking (Liu et al., 2019), meaning each example is masked differently in each epoch.

#### 4.2.2 Predictor Models

We use Gemma-2b (Team et al., 2024) as the predictor model. We fine-tune the raw language model for the three datasets (referred to as gemma-ft) using Low-Rank Adaptation (LoRA). Additionally, we employ an off-the-shelf instruct-tuned version (gemma-it) for zero-shot evaluation. We then conduct a detailed comparison between these two versions—fine-tuned (gemma-ft) and non-fine-tuned (gemma-it)—to assess their differences in attribution evaluation.

### 4.3 Attribution Methods

Here we detail the six widely used attribution methods employed in our study. We use all attribution methods in a contrastive way (Yin and Neubig, 2022). Contrastive attributions measure which features from the input make the foil token $y_f$ more likely and the target token $y_t$ less likely. We denote contrastive, target, and foil attributions by $S^C$, $S^t$, and $S^f$ respectively:

$$S^C = S^t - S^f \tag{1}$$

We use the implementation of these attribution methods provided by Yin and Neubig (2022) (for Gradient × input, gradient norm and erasure) and by Captum (Miglani et al., 2023) (for KernelSHAP and Integrated Gradient).

#### 4.3.1 Gradient Norm

We can calculate attributions based on the norm of the gradient of the model's prediction with respect to the input $x$ (Simonyan et al., 2013; Li et al., 2016).The gradient with respect to feature $x_i$ is

given by:

$$g(x_i) = \nabla_{x_i} q(y_t|x)$$

Where $q(y_t|x)$ is the model output for token $y_t$ given the input $x$. The contrastive gradient:

$$g^C(x_i) = \nabla_{x_i} \left( q(y_t|\boldsymbol{x}) - q(y_f|\boldsymbol{x}) \right)$$

We will use both norm one (gradnorm1) and norm two (gradnorm2):

$$S^C_{GN1}(x_i) = ||g^C(x_i)||_{L1}$$

$$S^C_{GN2}(x_i) = ||g^C(x_i)||_{L2}$$

### 4.3.2 Gradient $\times$ Input

In gradient $\times$ input (gradinp) method (Shrikumar et al., 2016; Denil et al., 2014), we compute the dot product of the gradient and the input token embedding $x_i$:

$$S_{GI}(x_i) = g(x_i) \cdot x_i$$

By multiplying the gradient by the input embedding, we also account for how much each token is expressed in the attribution score. The Contrastive Gradient $\times$ Input is:

$$S^C_{GI}(x_i) = g^C(x_i) \cdot x_i$$

### 4.3.3 Erasure

Erasure-based methods measure the importance of each token by erasing it and observing the effect on the model output (Li et al., 2017). This is achieved by taking the difference between the model output with the full input $x$ and the model output with the input where token $x_i$ is zeroed out, denoted as $x_{\neg i}$:

$$S^t_E(x_i) = q(y_t|x) - q(y_t|x_{\neg i})$$

For the contrastive case, $S^C_E(x_i)$ becomes:

$$\left( q(y_t|x) - q(y_t|x_{\neg i}) \right) - \left( q(y_f|x) - q(y_f|x_{\neg i}) \right)$$

### 4.3.4 KernelSHAP

KernelSHAP (Lundberg and Lee, 2017) explains the prediction of a classifier $q$ by learning a linear model $\phi$ locally around each prediction. The objective function of KernelSHAP constructs an explanation that approximates the behavior of $q$ accurately in the neighborhood of $x$. More important features have higher weights in this linear model $\phi$. Let $Z$ be a set of $N$ randomly sampled perturbations around $x$:

$$S^t_\phi = \arg\min_\phi \sum_{z \in Z} [q(y_t|z) - \phi^T z]^2 \pi_x(z) \quad (2)$$

KernelSHAP uses a kernel $\pi_x$ that satisfies certain principles when input features are considered agents of a cooperative game in game theory. We use equation 2 in a contrastive way. First we normalize $S^t_\phi$ and $S^f_\phi$ by dividing by their $L2$ norm and then subtracting:

$$S^C_\phi = \frac{S^t_\phi}{||S^t_\phi||} - \frac{S^f_\phi}{||S^f_\phi||} \quad (3)$$

### 4.3.5 Integrated Gradients

Integrated Gradients (IG) (Sundararajan et al., 2017) is a gradient-based method which addresses the problem of saturation: gradients may get close to zero for a well-fitted function. IG requires a baseline $\mathbf{b}$ as a way of contrasting the given input with the absence of information. For input $i$, we compute:

$$S^t_{IG}(x_i) = \frac{1}{m} \sum_{k=1}^{m} \nabla_{x_i} q\left( y_t \Big| b + \frac{k}{m}(x-b) \right) \cdot (x_i - b_i) \quad (4)$$

That is, we average over $m$ gradients, with the inputs to $q$ being linearly interpolated between the baseline $b$ and the original input $x$ in $m$ steps. We then take the dot product of that averaged gradient with the input embedding $\mathbf{x}_i$ minus the baseline.

We use a zero vector baseline (Mudrakarta et al., 2018) and five steps. The contrastive case becomes:

$$S^C_{IG} = \frac{S^t_{IG}}{||S^t_{IG}||} - \frac{S^f_{IG}}{||S^f_{IG}||} \quad (5)$$

## 5 Results and Discussion

### 5.1 The Out-of-Distribution Problem

**Why should we use counterfactuals instead of erasing important tokens or replacing them with unimportant ones?** First, we demonstrate that our counterfactual generators produce in-distribution text for the predictor. Second, we show that the rankings of attribution methods' faithfulness are consistent when using a counterfactual generator for token replacement, but these rankings differ when other replacement methods are used.

To achieve our first goal—demonstrating that the generated counterfactuals are in-distribution—we employ an out-of-distribution (OOD) detection

| Editor | gradnorm1 | | Erasure | | KernelSHAP | |
|---|---|---|---|---|---|---|
| | gemma-ft | gemma-it | gemma-ft | gemma-it | gemma-ft | gemma-it |
| pythia1 (ours) | 1.1 | 1.4 | 1.3 | 1.6 | 0.7 | 1.7 |
| pythia2 (ours) | 0.4 | 2.6 | 0.9 | 1.3 | 0.8 | 2.3 |
| gptj (ours) | 0.7 | 8.3 | 2.0 | 10.9 | 0.9 | 6.4 |
| erase | 0.3 | 19.9 | 2.3 | 32.8 | 0.6 | 81.4 |
| unk | 0.6 | 97.5 | 1.8 | 97.3 | 1.3 | 99.8 |
| mask | 0.0 | 94.8 | 0.5 | 93.3 | 0.0 | 98.5 |
| att-zero | 0.1 | 80.9 | 0.1 | 62.6 | 0 | 74.1 |

Table 1: OOD percentage when our counterfactual editor models generate samples, compared to other replacement methods (erase, unk, mask, and att-zero methods). This represents the percentage of corrupted examples that fall outside the 99th percentile of the NLL of the original sentences in the SST-2 dataset (lower is better). Scenarios with very high OOD percentages are highlighted.

| Attribution method | SST-2 | | | IMDB | | | AG-News | | |
|---|---|---|---|---|---|---|---|---|---|
| | pythia1 | pythia2 | gptj | pythia1 | pythia2 | gptj | pythia1 | pythia2 | gptj |
| gradnorm1 | 33.5 | **34.8** | **32.2** | **29.1** | **30.3** | 32.4 | 42.9 | 45.1 | 44.0 |
| gradnorm2 | **33.4** | 35.6 | 32.6 | 31.0 | 30.5 | **32.4** | 42.6 | 44.4 | 43.9 |
| gradinp | 40.5 | 41.8 | 40.8 | 36.1 | 36.3 | 36.5 | 43.1 | 44.6 | **42.2** |
| erasure | 35.5 | 36.6 | 33.4 | 32.7 | 32.7 | 34.4 | **42.0** | **42.7** | 43.0 |
| IG | 45.7 | 45.8 | 43.7 | 43.3 | 44.3 | 42.5 | 43.8 | 46.7 | 44.0 |
| KernelSHAP | 44.1 | 45.9 | 44.9 | 44.0 | 43.3 | 44.2 | 44.0 | 46.5 | 44.3 |
| Random | 44.6 | 46.0 | 44.3 | 43.8 | 42.7 | 43.2 | 44.0 | 46.0 | 44.0 |

Table 2: The mean percentage of tokens needed to be masked to achieve flipping Gemma-ft's label or reaching 50 percent masking in 200 examples from evaluation split of SST-2, IMDB, and AG-News datasets (lower is better). pythia1, pythia2, and gptj models are used to fill the masks and generate counterfactuals.

technique to measure the percentage of our generated inputs that are OOD. Prominent OOD detection methods use a threshold, considering any input with a value higher than this threshold as OOD (Chen et al., 2023). For each dataset, we calculate the threshold by measuring the negative log-likelihood (NLL) of 200 original examples using different predictors (fine-tuned and instruct-tuned) and consider the 99th percentile of these NLLs as the OOD threshold. We use NLL to detect OOD because the type of shift we aim to detect is background shift. OOD data can be classified as either semantic or background shift (Arora et al., 2021). Semantic features have a strong correlation with the label, and semantic shift occurs when we encounter unseen classes at test time. In contrast, background features consist of population-level statistics that do not depend on the label and focus on the style of the text.

In evaluating faithfulness by corrupting the input, we do not introduce new labels or classes; instead,

we change the style of the text. Therefore, we aim to detect background shift. There are two common types of OOD detection methods: calibration and density estimation. Density estimation methods, such as perplexity (PPL), outperform calibration methods under background shifts, while the opposite is true under semantic shift. We use NLL, which is closely related to PPL.

An attribution method shows us which tokens are important, and we replace those tokens in four ways: (i) using an editor to replace the tokens (our method), (ii) using tokens that are considered semantically unimportant (the <unk> token and the <mask> token), (iii) erasing the tokens, and (iv) zeroing out the attention mask for important tokens without altering the text itself (att-zero).

The baselines (ii) through (iv) are similar to previous work (Hase et al., 2021). Table 1 shows that for both fine-tuned and instruct-tuned predictors, the generated counterfactuals are mostly in-distribution. Specifically, we present results for

| Attribution method | SST-2 | | | IMDB | | | AG-News | | |
|---|---|---|---|---|---|---|---|---|---|
| | pythia1 | pythia2 | gptj | pythia1 | pythia2 | gptj | pythia1 | pythia2 | gptj |
| gradnorm1 | 41.4 | 42.0 | **40.3** | 42.1 | 42.3 | 44.0 | 46.6 | 46.8 | 40.0 |
| gradnorm2 | 41.5 | 42.2 | 40.6 | 42.3 | 42.6 | 43.6 | 46.6 | 46.9 | 39.9 |
| gradinp | 42.9 | 43.4 | 43.2 | **41.2** | **41.4** | **42.2** | 45.8 | 45.3 | 39.2 |
| erasure | **40.8** | **41.5** | 41.0 | 43.1 | 42.8 | 43.8 | 45.0 | 45.5 | 39.5 |
| IG | 44.7 | 44.0 | 45.4 | 43.4 | 43.2 | 44.0 | 45.7 | 45.3 | 38.6 |
| KernelSHAP | 43.6 | 43.5 | 44.0 | 43.7 | 42.9 | 43.9 | 46.1 | 45.3 | **37.2** |
| Random | 44.8 | 44.7 | 44.3 | 45.4 | 46.1 | 45.8 | **44.9** | **45.2** | 39.2 |

Table 3: The mean percentage of tokens needed to be masked to achieve flipping Gemma-it's label or reaching 50 percent masking in 200 examples from evaluation split of SST-2, IMDB, and AG-News datasets (lower is better). pythia1, pythia2, and gptj models are used to fill the masks and generate counterfactuals.

the SST-2 dataset and three attribution methods; results for other attribution methods and datasets are shown in Appendix A. Each number in Table 1 represents the average over five levels of replacement (10% to 50%) and 200 examples from evaluation sets.

Chen et al. (2023) demonstrate that fine-tuning renders a model insensitive to non-semantic shifts. Their research indicates that fine-tuning eliminates pre-trained, task-agnostic knowledge about general linguistic properties, which is crucial for detecting non-semantic shifts. Our findings align with these observations. When a predictor is fine-tuned for a specific classification task, such as sentiment analysis on the SST-2 dataset, it is optimized to assign high probabilities to the correct labels for the training data. Consequently, this fine-tuned model becomes less sensitive to input corruptions. In our experiments, regardless of the replacement method employed, the resulting inputs tend to remain in-distribution for the fine-tuned predictors. As evidenced in Table 1, under the Gemma-ft columns, the percentage of out-of-distribution (OOD) examples approaches zero.

In contrast, Gemma-it, an off-the-shelf model that is not optimized for a specific dataset, exhibits different behavior. When subjected to various input modifications—such as replacing important tokens with semantically neutral ones (e.g., <unk> or <mask> tokens), completely removing tokens, or zeroing out the attention mask for important tokens without altering the text itself—the Gemma-it predictor frequently categorizes these modified inputs as OOD. This disparity in behavior between fine-tuned and off-the-shelf models underscores the impact of task-specific optimization on a model's

sensitivity to input perturbations. However, when the counterfactual generator is used to modify the inputs, the examples remain in-distribution even for the instruct-tuned predictor. This observation demonstrates that when we do not want to change the predictor model and prefer to use an off-the-shelf model as our predictor, using a counterfactual generator is helpful in evaluating the faithfulness of attribution methods.

To achieve our second goal—demonstrating the consistency of the faithfulness rankings of attribution methods when using a counterfactual generator, and the lack of consistency when another replacement method is applied—we use Spearman's rank correlation, as in previous works (Rong et al., 2022). For each example, we rank the attribution methods based on the percentage of the mask needed to flip the label. We then compute the correlations among these rankings across all seven replacement methods (our three editors, Erase, <unk>, <mask>, and att-zero) and average the results over 200 examples.

We present this analysis for the SST-2 dataset in Figure 4. Other datasets yield similar results and are shown in Appendix B. In the top correlation matrix of Figure 4, these average correlations are shown for the fine-tuned predictor. For the fine-tuned predictor, all replacement methods have high average correlations with each other. The middle matrix in Figure 4 shows these correlations when the predictor model is an off-the-shelf instruct-tuned model. For the off-the-shelf predictor, only when a counterfactual generator is used do the rankings have high correlations with each other; other replacement methods have low correlations with the counterfactual generators. This is

likely because, when using an instruct-tuned predictor, replacement methods other than counterfactual generators create OOD inputs.

The bottom matrix of Figure 4 displays the difference between the first and second matrices. It shows that the correlation difference between fine-tuned and instruct-tuned predictors is near zero when using editors as the replacement method. However, the difference is significant when using other replacement methods (<unk>/Erase/<mask>/att-zero). This suggests that when evaluating explanations on an off-the-shelf instruct-tuned model, it is crucial to avoid using corrupted OOD text.

## 5.2 Analysis of Feature Importance Methods

In Tables 2 and 3, we show the average masking percentage required (the average percentage of tokens the counterfactual generator should change) to flip the label for fine-tuned and instruct-tuned predictor models, respectively. The masking percentage is highly correlated with the flip rate—the percentage of labels each counterfactual generator is able to flip by altering the corrupted tokens. In Appendix C, we show the flip rate for both fine-tuned and instruct-tuned predictor models. Attribution methods that can flip the labels with less masking (i.e., fewer changes) are also able to flip more labels.

For the fine-tuned predictor (Table 2), gradient norm methods consistently outperform others on the SST-2 and IMDB datasets. In contrast, for AG-News, the Erasure method consistently performs the best or near the best. Our results suggest that straightforward methods, such as gradnorm1, gradnorm2, and Erasure, consistently deliver superior performance regardless of the editor used.

For the instruct-tuned predictor (Table 3), the Erasure method yields the best results for the SST-2 dataset, while gradinp demonstrates the best performance on the IMDB dataset. However, no attribution method consistently outperforms random selection for the AG-News dataset. Overall, these findings suggest that attribution methods are less effective when the model is not fine-tuned for the specific task, indicating the need for cautious application of these methods to pretrained and instruct-tuned language models.

## 6 Conclusion

In this work, we designed a faithfulness evaluation protocol based on counterfactual generation. We demonstrated that the efficacy of attribution methods varies between models fine-tuned on our specific dataset and off-the-shelf, instruct-tuned models. We showed that counterfactual generators are effective for evaluating feature attribution because they can produce mostly in-distribution text for the predictor model. This approach allows us to separate the evaluation of the model from the evaluation of the attribution method, as the examples used are mostly in-distribution. We also found high consistency between different counterfactual generators and a lack of consistency with other replacement methods, highlighting the importance of being in-distribution, particularly when evaluating attributions on off-the-shelf models. Finally, we used our protocol to compare different attribution methods.

## 7 Limitations

Our work is limited in several aspects: First, we rely on generating counterfactuals, which requires a strong generative model. Generating counterfactuals—especially for long sequences—is computationally expensive. Second, the counterfactual generator might unintentionally incorporate the artifacts and shortcuts used by the predictor to flip the label, potentially limiting the intended application of our approach. Third, we applied our protocol only to classification tasks; evaluating it on other tasks, like translation, is left for future work.

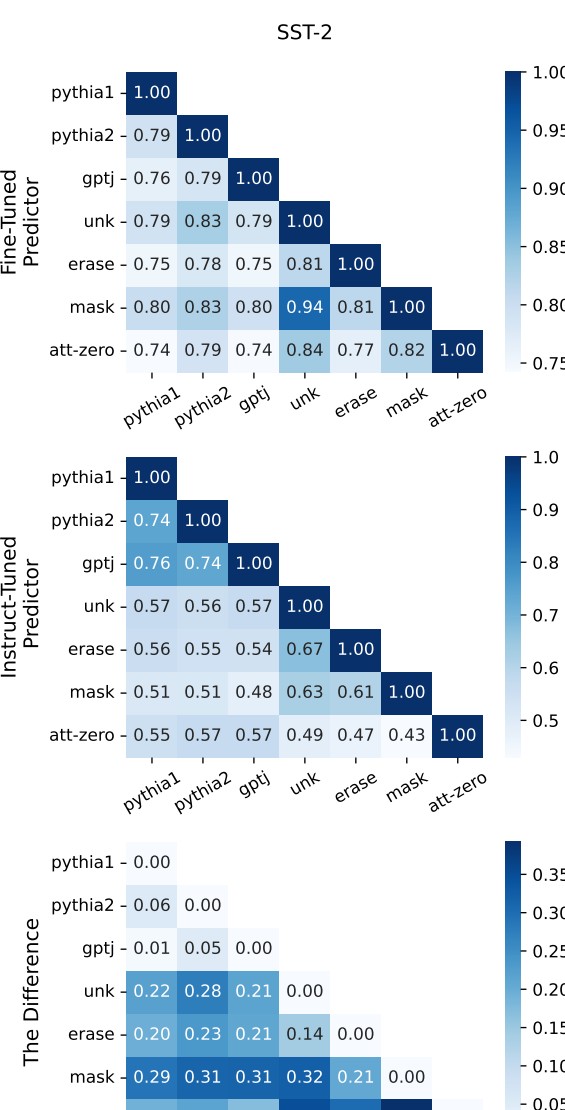

Figure 4: The top matrix presents the average correlation of attribution ranks for the fine-tuned predictor. The middle matrix shows the average correlation of attribution ranks when using an off-the-shelf instruct-tuned predictor. The bottom matrix illustrates the difference between the fine-tuned and instruct-tuned models, indicating that when editors are used as the replacement method, the difference in correlation is near zero. In contrast, using other replacement methods (i.e., <unk>, erase, <mask>, att-zero) results in significant inconsistencies between the two predictor types, likely due to the creation of out-of-distribution (OOD) text for the instruct-tuned model.

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

# A

Table 4 is the OOD percentages for other attribution methods in SST-2 dataset that were not in Table 1. Tables 5 and 6 show OOD percentages in AG-News dataset.

# B

Figures 5 and 6 show the difference of correlations in IMDB and AG-News datasets respectively.

# C

Tables 7 and 8 show flip-rates for fine-tuned and instruct-tuned predictor models respectively.

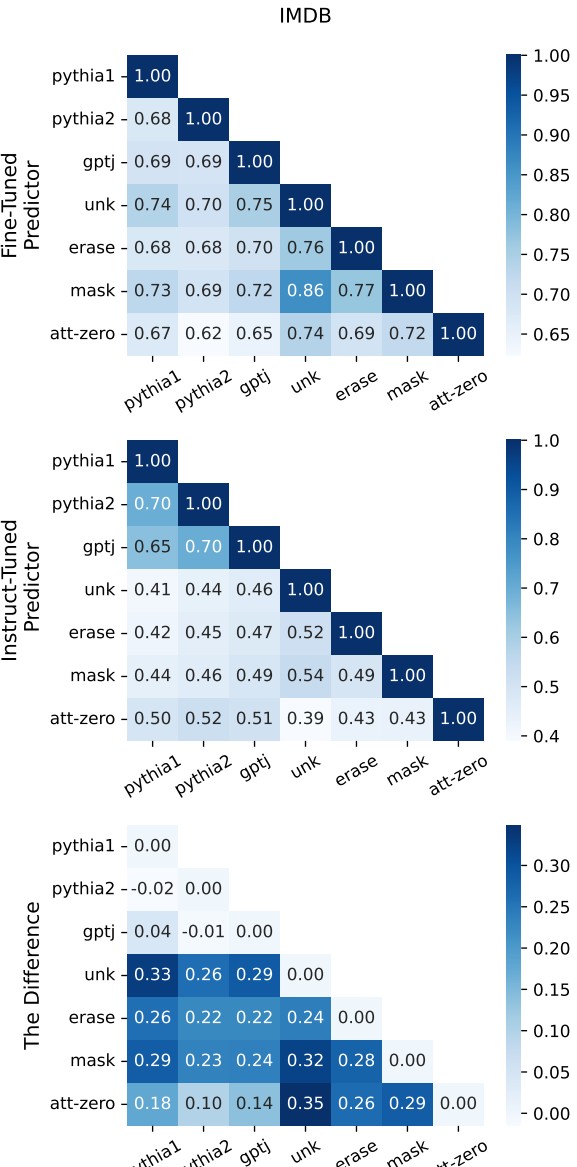

Figure 5: The difference

| Editor | gradnorm2 | | gradinp | | integrated gradient | |
|---|---|---|---|---|---|---|
| | gemma-ft | gemma-it | gemma-ft | gemma-it | gemma-ft | gemma-it |
| pythia1 (ours) | 1.4 | 1.2 | 1.3 | 1.5 | 1.1 | 0.7 |
| pythia2 (ours) | 0.4 | 2.6 | 0.8 | 1.6 | 1.0 | 0.8 |
| gptj (ours) | 0.8 | 8.2 | 2.4 | 5.2 | 0.4 | 8.7 |
| erase | 0.5 | 21.9 | 1.1 | 83.8 | 1.4 | 77.4 |
| unk | 0.6 | 98.4 | 1.6 | 100.0 | 3.4 | 99.3 |
| mask | 0.0 | 95.8 | 0.2 | 99.5 | 0.4 | 97.8 |
| att-zero | 0.1 | 80.8 | 0.0 | 70.6 | 0 | 65.8 |

Table 4: OOD percentage when our counterfactual editor models generate samples, compared to other replacement methods (erase, unk, mask, and att-zero methods). This is the percentage of corrupted examples that are out of the 99th percentile of the NLL of the original sentences in SST-2 dataset (lower is better). The scenarios with very high numbers of OODs are highlighted.

| Editor | gradnorm1 | | Erasure | | KernelSHAP | |
|---|---|---|---|---|---|---|
| | gemma-ft | gemma-it | gemma-ft | gemma-it | gemma-ft | gemma-it |
| pythia1 (ours) | 2.4 | 5.0 | 4.1 | 4.9 | 1.6 | 4.5 |
| pythia2 (ours) | 2.4 | 5.2 | 3.1 | 4.9 | 2.0 | 6.9 |
| gptj (ours) | 1.0 | 5.0 | 1.5 | 4.9 | 0.9 | 7.8 |
| erase | 3.5 | 11.5 | 8.8 | 46.3 | 2.3 | 74.9 |
| unk | 1.0 | 98.2 | 3.8 | 99.7 | 1.3 | 99.9 |
| mask | 0.7 | 85.7 | 5.2 | 96.7 | 0.5 | 98.2 |
| att-zero | 4.9 | 79.1 | 3.2 | 57.4 | 0.8 | 62.6 |

Table 5: OOD percentage when our counterfactual editor models generate samples, compared to other replacement methods (erase, unk, mask, and att-zero methods). This is the percentage of corrupted examples that are out of the 99th percentile of the NLL of the original sentences in AG-News dataset (lower is better). The scenarios with very high numbers of OODs are highlighted.

| Editor | gradnorm2 | | gradinp | | integrated gradient | |
|---|---|---|---|---|---|---|
| | gemma-ft | gemma-it | gemma-ft | gemma-it | gemma-ft | gemma-it |
| pythia1 (ours) | 2.2 | 5.0 | 1.4 | 4.5 | 1.6 | 4.5 |
| pythia2 (ours) | 2.4 | 5.2 | 2.3 | 5.5 | 1.3 | 5.9 |
| gptj (ours) | 1.2 | 5.0 | 2.0 | 5.1 | 1.2 | 6.4 |
| erase | 3.3 | 11.6 | 2.6 | 60.3 | 1.9 | 55.5 |
| unk | 0.9 | 98.0 | 1.7 | 99.3 | 1.3 | 99.9 |
| mask | 0.8 | 86.4 | 1.5 | 94.1 | 0.5 | 98.2 |
| att-zero | 5.1 | 78.9 | 1.9 | 58.7 | 0.6 | 50.8 |

Table 6: OOD percentage when our counterfactual editor models generate samples, compared to other replacement methods (erase, unk, mask, and att-zero methods). This is the percentage of corrupted examples that are out of the 99th percentile of the NLL of the original sentences in AG-News dataset (lower is better). The scenarios with very high numbers of OODs are highlighted.

| Attribution method | SST-2 | | | IMDB | | | AG-News | | |
|---|---|---|---|---|---|---|---|---|---|
| | pythia1 | pythia2 | gptj | pythia1 | pythia2 | gptj | pythia1 | pythia2 | gptj |
| gradnorm1 | 67.5 | 63.5 | 72.5 | 78.5 | 76.5 | 68.8 | 22.0 | 18.5 | 17.0 |
| gradnorm2 | 66.5 | 61.0 | 71.0 | 75.5 | 80.5 | 67.5 | 22.0 | 19.5 | 17.0 |
| gradinp | 32.0 | 31.0 | 33.0 | 54.5 | 50.5 | 49.0 | 21.5 | 21.0 | 23.0 |
| erasure | 56.5 | 47.5 | 56.5 | 59.0 | 61.5 | 55.5 | 24.5 | 24.0 | 23.0 |
| IG | 18.5 | 17.5 | 23.5 | 33.5 | 31.5 | 41.0 | 16.5 | 14.5 | 21.0 |
| KernelSHAP | 22.0 | 17.5 | 22.5 | 30.5 | 34.0 | 31.0 | 18.0 | 13.5 | 18.5 |
| Random | 22.5 | 18.5 | 23.5 | 33.5 | 38.5 | 35.0 | 18.5 | 15.5 | 18.5 |

Table 7: The mean percentage of success in flipping Gemma-ft's label in 200 examples of evaluation split in SST-2, IMDB, and AG-News datasets (higher is better).

| Attribution method | SST-2 | | | IMDB | | | AG-News | | |
|---|---|---|---|---|---|---|---|---|---|
| | pythia1 | pythia2 | gptj | pythia1 | pythia2 | gptj | pythia1 | pythia2 | gptj |
| gradnorm1 | 34.5 | 31.0 | 33.5 | 37.5 | 36.0 | 44.0 | 12.0 | 10.0 | 18.0 |
| gradnorm2 | 34.0 | 31.5 | 33.5 | 37.0 | 35.0 | 43.5 | 12.0 | 10.0 | 18.5 |
| gradinp | 27.0 | 25.0 | 28.0 | 38.5 | 36.5 | 62.0 | 18.5 | 18.5 | 17.0 |
| erasure | 29.5 | 26.5 | 32.5 | 29.5 | 26.5 | 46.0 | 17.0 | 16.0 | 21.5 |
| IG | 23.5 | 24.5 | 23.0 | 37.0 | 38.0 | 61.0 | 17.5 | 18.5 | 12.0 |
| KernelSHAP | 28.0 | 26.0 | 21.5 | 34.0 | 36.0 | 59.5 | 17.5 | 18.0 | 19.0 |
| Random | 21.0 | 23.0 | 18.5 | 30.0 | 30.5 | 55.5 | 20.5 | 19.0 | 17.0 |

Table 8: The mean percentage of success in flipping Gemma-it's label in 200 examples of evaluation split in SST-2, IMDB, and AG-News datasets (higher is better).

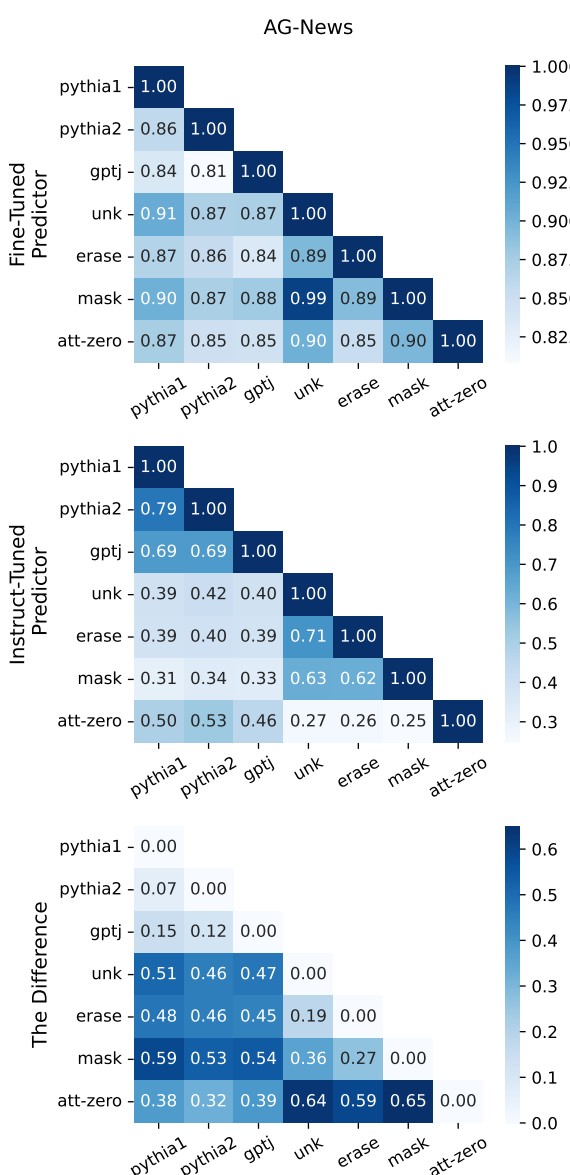

Figure 6: The difference