# OpenReview forum: "Counterfactuals As a Means for Evaluating Faithfulness of Attribution Methods in Autoregressive Language Models"
_EMNLP/2024/Workshop/BlackBoxNLP — BlackboxNLP 2024_

### Official Review · Reviewer_Kqp4 · 2024-09-09

**Overall Assessment:** 3
**Confidence:** 3

**Best Paper:**

1

**Best Paper Justification:**

N/A

**Comments Questions Suggestions And Typos:**

* Parenthetical and inline citations are used incorrectly throughout the text. E.g., in line 293 you should have a parenthetical citation: “We use Gemma-2b Team et al. (2024) as the predictor.”
* In sections 4.3.1 and 4.3.2, the FI methods are given as a function of a specific feature $x_i$ (which I believe represents each token in the input). Sections 4.3.3, 4.3.4 and 4.3.5 do not have an $x_i$ input, however. Further, how per-token scores are computed for 4.3.4 is not clear in my opinion.

**Paper Summary:**

This paper proposes a new method to evaluate the faithfulness of feature importance (FI) methods.
Given a **predictor model** to be analysed and an input $x$, first run a feature importance method on it; this process outputs a list of which tokens in $x$ are most important for a prediction (and in which order).
Second, mask the top 10% tokens in $x$, and use a second **editor** model to create a counterfactual input $x’$ which should have an alternative label.
If the predictor model’s output changes on $x’$, stop; else, iteratively also mask the following top 10% tokens (according to the FI method), until labels change or 50% of tokens have been masked.
A better FI method should require changing less tokens to  flip the predictor model’s predictions.


Editor model training. Given a dataset of inputs + labels $(x, y)$, mask from 5 to 50% of the tokens in $x$, than create a sequence `mask(x) y <counterfactual> x`  and train an autoregressive editor model to reconstruct $x$ this way. This autoregressive model is initialised from a pretrained model, such as Pythia.

The paper evaluates their faithfulness evaluation protocol against prior methods using two experiments:
* The number of out-of-distribution inputs these faithfulness evaluation protocols generate. Their method generates less out-of-distribution inputs than other methods, at least when analysing instruction tuned models (results for fine-tuned models are mixed).
* The correlation between the rankings output by different faithfulness evaluation protocols. They show that their method, when using an editor model based on different pretrained models, leads to rankings which correlate with one another. Prior methods do not correlate among each other.

The paper then evaluates a number of FI methods using their metric, showing that simple methods—such as gradient norm—often over perform more complex methods.

**Summary Of Strengths:**

This paper tackles an interesting problem: how to evaluate the faithfulness of FI methods.

The paper points out an issue with prominent methods for faithfulness evaluation, and proposes an effective method to mitigate it.

The paper compares the percentage of out-of-distribution examples their method produces against baselines, and evaluates a number of FI methods.

**Summary Of Weaknesses:**

One way in which the paper evaluates their method is by showing the correlation between rankings using different faithfulness evaluation protocols. However, I do not think this is sufficiently motivated. Why should a high correlation in this experiment represent a better method? To my understanding, a high correlation between rankings when starting the editor model with different pretrained models only shows that the method is robust to this choice, but not necessarily that this is a better method.

The paper evaluates the presence of out-of-distribution inputs according to their negative likelihood. However, to the best of my knowledge, likelihood only correlates weakly with text quality and how out-of-distribution it is (Holtzman et al. 2020). Running human evaluations could increase confidence in these results.


* Holtzman et al. (2020). The Curious Case of Neural Text Degeneration

---

### Official Review · Reviewer_Xcib · 2024-09-09

**Overall Assessment:** 4
**Confidence:** 4

**Best Paper:**

1

**Best Paper Justification:**

.

**Comments Questions Suggestions And Typos:**

- "We use counterfactual generators to change the input, focusing on the important tokens specified by the attribution methods, and make sure that the input to the model is natural, fluent, and in-distribution." How do you make sure?
- "We argue that if an attribution method helps the counterfactual generator to change the model’s prediction with fewer changes, that method knows more about the model’s inner workings, which means it is more faithful." I think this needs a justification.
- Row 242, a fragment?

**Paper Summary:**

In this manuscript, the authors propose an algorithm to evaluate explanations of LLMs by leveraging datasets of counterfactuals. Specifically, the algorithm fits a counterfactual generator model which, when given an instance and a mask of possible tokens to perturb, attempts to generate counterfactual by perturbing said mask. The higher the success rate, the higher the importance of said mask.

**Summary Of Strengths:**

1. High novelty
2. In-distribution evaluation

**Summary Of Weaknesses:**

1. Some claims ought to be better justified, see observations below

---

### Official Review · Reviewer_5Cwj · 2024-09-11

**Overall Assessment:** 3
**Confidence:** 3

**Best Paper:**

1

**Best Paper Justification:**

NA

**Comments Questions Suggestions And Typos:**

Line 179 attributions -> Attributions
Line 132 hence -> Hence
Line 71 wrong citation format for Hase et al. 2021.

**Paper Summary:**

This paper expands the traditional faithfulness evaluation method for attribution with a counterfactual generator. The authors argue that the current faithfulness evaluation often involves text replacement methods that create OOD inputs. The counterfactual generator ensures in-distribution input, and could help to boost the consistency across different attribution methods.

**Summary Of Strengths:**

1. This paper proposes a novel solution to OOD problem frequently encountered in traditional masking or erasing approaches used in attribution methods.
2. The paper provides a thorough evaluation to several popular attribution methods, including gradient norm, gradient × input, Erasure, KernelSHAP, and integrated gradient, proving the generalizability of the proposed method.

**Summary Of Weaknesses:**

1. The literature regarding counterfactual generation and evaluation in LLMs is missing:
e.g. Qin, L., Bosselut, A., Holtzman, A., Bhagavatula, C., Clark, E., & Choi, Y. (2019). Counterfactual story reasoning and generation. arXiv preprint arXiv:1909.04076.
Meng, K., Bau, D., Andonian, A., & Belinkov, Y. (2022). Locating and editing factual associations in GPT. Advances in Neural Information Processing Systems, 35, 17359-17372.
Elazar, Y., Ravfogel, S., Jacovi, A., & Goldberg, Y. (2021). Amnesic probing: Behavioral explanation with amnesic counterfactuals. Transactions of the Association for Computational Linguistics, 9, 160-175.
Li, J., Yu, L., & Ettinger, A. (2023). Counterfactual reasoning: Testing language models' understanding of hypothetical scenarios. arXiv preprint arXiv:2305.16572.

2. I am concerned about the quality of the text generated by counterfactual generator. For instance, the most intuitive counterfactual by masking “clear” in "a clear sense of purpose" would be "an unclear sense of purpose", but the current generator might create text like "a purposeful sense of purpose". If I am understanding it correctly, the current counterfactual generator shown in Fig 3 seems to be more a like a conventional unmasker, the training example does not have a contrastive original-counterfactual pair, and the only hint comes from the token "<counterfactual>".

3. I am not sure if I have fully understand the editor model in the attribution evaluation process. Does it only perform masked token prediction, or does it jointly predict the label for generated counterfactual texts? I assume that the label for the generated counterfactuals would also be predicted, but it is not clear to me where this step is being done from Fig 2.

---

### Decision · Program_Chairs · 2024-09-20

**Decision:**

Accept

**Comment:**

The paper introduces a novel method for evaluating the faithfulness of feature attribution methods for language models, based on a incremental masking procedure that measures the amount of tokens that need to be corrupted in order to flip a model prediction. The reviewers commend the novelty of this method, as well as its evaluation with respect to other well known attribution methods. The authors are encouraged to incorporate a reflection of the weaknesses pointed out by reviewers 5Cwj and Kqp4 in the revised version of the paper.